# Sialic Acid-Containing Glycans as Cellular Receptors for Ocular Human Adenoviruses: Implications for Tropism and Treatment

**DOI:** 10.3390/v11050395

**Published:** 2019-04-27

**Authors:** Naresh Chandra, Lars Frängsmyr, Sophie Imhof, Rémi Caraballo, Mikael Elofsson, Niklas Arnberg

**Affiliations:** 1Section of Virology, Department of Clinical Microbiology, Umeå University, SE-90185 Umeå, Sweden; naresh.chandra@umu.se (N.C.); lars.frangsmyr@umu.se (L.F.); imhofsophie4@gmail.com (S.I.); 2Department of Chemistry, Umeå University, SE-90187 Umeå, Sweden; remi.caraballo@umu.se (R.C.); mikael.elofsson@umu.se (M.E.)

**Keywords:** adenovirus, sialic acid, cellular receptor, pharyngoconjunctival fever, epidemic keratoconjunctivitis, tropism

## Abstract

Human adenoviruses (HAdV) are the most common cause of ocular infections. Species B human adenovirus type 3 (HAdV-B3) causes pharyngoconjunctival fever (PCF), whereas HAdV-D8, -D37, and -D64 cause epidemic keratoconjunctivitis (EKC). Recently, HAdV-D53, -D54, and -D56 emerged as new EKC-causing agents. HAdV-E4 is associated with both PCF and EKC. We have previously demonstrated that HAdV-D37 uses sialic acid (SA)-containing glycans as cellular receptors on human corneal epithelial (HCE) cells, and the virus interaction with SA is mediated by the knob domain of the viral fiber protein. Here, by means of cell-based assays and using neuraminidase (a SA-cleaving enzyme), we investigated whether ocular HAdVs other than HAdV-D37 also use SA-containing glycans as receptors on HCE cells. We found that HAdV-E4 and -D56 infect HCE cells independent of SAs, whereas HAdV-D53 and -D64 use SAs as cellular receptors. HAdV-D8 and -D54 fiber knobs also bound to cell-surface SAs. Surprisingly, HCE cells were found resistant to HAdV-B3 infection. We also demonstrated that the SA-based molecule i.e., ME0462, designed to bind to SA-binding sites on the HAdV-D37 fiber knob, efficiently prevents binding and infection of several EKC-causing HAdVs. Surface plasmon resonance analysis confirmed a direct interaction between ME0462 and fiber knobs. Altogether, we demonstrate that SA-containing glycans serve as receptors for multiple EKC-causing HAdVs, and, that SA-based compound function as a broad-spectrum antiviral against known and emerging EKC-causing HAdVs.

## 1. Introduction

To date, 90 human adenovirus (HAdV) types have been identified and are classified into seven species (A–G) [1]. More than half of these types belong to species D HAdV (HAdV-D), including a number of viruses of recombinant origins. HAdVs are associated with infections in the airway (species A, B, C, and E), gut (species F and G), and eyes (species B, D, and E) [2]. These infections are mostly self-limiting, however, they can be life-threatening in individuals with compromised immune systems [2].

HAdVs are the major cause of infectious conjunctivitis, constituting up to 75% of all conjunctivitis cases worldwide [3,4]. It is estimated that each year around 20–30 million individuals suffer from HAdV-associated conjunctivitis globally [4,5,6]. In Japan alone, HAdV-associated conjunctivitis affects approximately one million individuals each year [5,6]. Selected members of species B, D, and E HAdVs cause conjunctivitis with a broad range of severity [7]. Conjunctivitis caused by species B HAdV type 3 (HAdV-B3) and HAdV-E4 is referred to as pharyngoconjunctival fever (PCF), which primarily involves the infection of conjunctiva [3,7,8,9]. Individuals suffering from PCF display mild symptoms such as fever, pharyngitis, rhinitis, and follicular conjunctivitis. PCF is self-limiting and usually resolves within 2–3 weeks [4]. A limited number of species D HAdVs cause a more severe ocular infection, epidemic keratoconjunctivitis (EKC), which is usually restricted to the eye [3]. However, recently several EKC-causing HAdVs have also been detected in patients with urethritis [10]. Additionally, there are cases of mild forms of EKC caused by HAdV-E4 [3,4]. EKC is a severe eye infection and involves both conjunctiva and cornea [4]. Historically, HAdV-D8, -D37, and –D64 (previously known as HAdV-D19a) have been considered as major causes of EKC [3]. However, in recent years emerging HAdV types such as HAdV-D53, -D54, and –D56 have also been reported to cause EKC [3,5,6,11,12]. According to several recent reports, HAdV-D54 has become the leading cause of EKC in Japan [6,12,13]. Among all EKC-causing HAdVs, HAdV-D8 is associated with the most severe clinical manifestations, including full corneal layer detachment [14,15]. During the acute phase, EKC patients display symptoms such as redness of eyes, excessive tearing, foreign-body sensation, edema, lacrimation, and photophobia. Corneal cells infected by EKC-causing HAdVs secrete chemokines such as IL8 and MCP1, which induce the infiltration of various immune cells (subepithelial infiltrates) into the corneal stroma [3,4,16] and the presence of these subepithelial infiltrates in the stroma is a hallmark of EKC. These infiltrates can persist in the stroma from months to years and can lead to visual impairment [3,17]. EKC usually begins as a unilateral condition (infection of one eye), but in most cases, it becomes bilateral as a result of eye-to-hand-to-eye transmission [3]. Although EKC outbreaks are reported worldwide, they are frequently reported in densely populated areas/countries [5,18,19]. EKC-causing HAdVs are highly contagious, thus, patients suffering from EKC are advised to abstain from the workplace, which leads to big socio-economic loss, particularly in developing countries [16]. Currently, no approved antivirals are available for the treatment of HAdV-associated ocular infections and the treatment strategies are mostly directed towards limiting the severity of symptoms and reducing the inflammation [3].

HAdVs attach to host cells through the interaction between the knob domain of the viral fiber protein and cell-surface receptors [20]. PCF-causing HAdV-B3 and -E4 use desmoglein 2 (DSG-2) and the coxsackie and adenovirus receptor (CAR) as cellular receptors, respectively [21,22]. These receptors were identified using cell lines of non-ocular origins. Given their ocular tropism, receptor usage by HAdV-B3 and -E4 on ocular cells has not been investigated. It has been shown that infection of corneal cells by acute hemorrhagic conjunctivitis (AHC) causing CVA24v is SA-dependent, whilst SAs do not function as receptors for CVA24v on HeLa cells [23]. Notably, another AHC-causing enterovirus 70 also uses SAs as cellular receptors [24]. Furthermore, HAdV-D37 uses SA-containing glycans as cellular receptors on human corneal epithelium (HCE) cells that represent the ocular tropism of EKC-causing HAdVs [25]. HAdV-D37 binds (via its highly positively charged fiber knob) to SA-containing hexasaccharide, resembling those in GD1a-gangliosides, and uses these glycans to infect HCE cells. HAdV-D8 and -D64 also utilize SA-containing glycans as cellular receptors on A549 cells [26,27], however, whether these viruses also use SAs as receptors on HCE cells is still unknown. Altogether, this highlights a potential link between SA-glycans and ocular viral pathogens. It has been suggested that the composition of SA-glycans and the expression of the types of SAs expressed on different cells can influence the receptor usage by viruses [23]. Thus, the possibility of HAdV-B3 and -E4 utilizing SA-containing receptors on ocular cells cannot be excluded. The cellular receptors for emerging EKC-causing HAdV-D53 and -D54 have not been characterized yet.

Here, by using HCE cells as a model cell line, we investigated the impact of cellular SA-containing glycans on the infection of ocular HAdVs. We demonstrate that SA-containing glycans are important for the infection of species D but not species E ocular HAdVs. We also evaluated the antiviral function of a SA-based molecule i.e., ME0462 (designated 17a in reference [26]), a known potent inhibitor of HAdV-D37 infection [28], against EKC-causing HAdVs.

## 2. Materials and Methods

### 2.1. Cells, Viruses, Antibodies, and Enzymes

Cells: Human corneal epithelial (HCE) cells (a gift from Dr. Araki-Sasaki) were grown in SHEM medium (1:1; DMEM (Dulbecco’s Modified Eagle Medium) and HAM’s F12 Nutrient Mixture supplemented with 20 mM 4-(2-hydroxyethyl)-1-piperazine-ethane-sulfonic-acid (HEPES), 5 μg/mL insulin, 0.5% dimethyl sulfoxide, 0.1 μg/mL cholera toxin, 10 ng/mL human epidermal growth factor (Sigma, St. Louis, MO, USA), 20 U/mL penicillin + 20 μg/mL streptomycin (PEST, Invitrogen, Carlsbad, CA, USA), and 10% fetal bovine serum (FBS; Sigma, St. Louis, MO, USA). A549 cells (a gift from Dr. Alistair Kidd) were grown in DMEM supplemented with 20 mM HEPES, PEST, and 10% FBS. Viruses: HAdV-B3, -E4, -D37, -D53, -D56, and -D64 viruses were produced in A549 cells as reported previously [29]. HAdV-D37, -D53, and -D64 were also produced with ^35^S-labeling. Viruses were eluted in phosphate buffer saline (PBS) on a NAP column (GE Healthcare, Chicago, IL, USA) and stored (at −20 °C) in PBS containing 10% glycerol. Antibodies: Monoclonal mouse anti-adenovirus antibody (clone 8052, Millipore, Burlington, NJ, USA) was used for infection experiments. Mouse monoclonal anti-RGS-His antibody (recognizes the epitope RGSHHHH; Qiagen, Hilden, Germany) was used for flow cytometry. Donkey anti-mouse Alexa Fluor 488 antibody (Invitrogen, Carlsbad, CA, USA) was used as a secondary antibody for infection and flow cytometry experiments. Neuraminidase (*Vibrio cholarae*) was purchased from Sigma (St. Louis, MO, USA).

### 2.2. Cloning, Expression, and Purification of Fiber Knobs

Cloning, expression, and purification of fiber knobs were carried out as described previously [30]. Briefly, HAdV-D8, -D37, and -D54 fiber knob genes were cloned into a pQE30-Xa expression vector encoding an N-terminal Histidine-tag (His-tag) using restriction sites for BamHI and XmaI (Thermo Scientific, Waltham, MA, USA). All constructs were confirmed by sequencing (Eurofins MWG Operon, Ebersberg, Germany). Fiber knobs were expressed in *Escherichia coli* (strain M15) according to the protocol from Qiagen (The QIAexpressionist; Qiagen, Hilden, Germany). Briefly, three liters of bacterial culture were incubated at 37 °C to an optical density of 0.6. The bacterial culture was then induced with 1 mM isopropyl β-d-1-thiogalactopyranoside (Thermo Scientific, Waltham, MA, USA). After 4‒5 h, the culture was centrifuged, pelleted, and stored at −20 °C. Fiber knobs were purified with Ni-NTA agarose beads followed by an anion exchange (Q-sepharose) chromatography.

### 2.3. Infection Assays

HCE cells were grown as a monolayer in the transparent flat bottom (30,000 cells/well) 96-well plates. The cells were then washed three times with serum-free growth medium. Prior to the infection experiment, we titrated the virus stocks and used dilutions resulting in infection of approximately 5% of cells in each well. HAdV-E4 (3000 vp/cell), -D37 (700 vp/cell), -D53 (550 vp/cell), -D56 (500 vp/cell), and -D64 (300 vp/cell) were added to cells and incubated for 1 h on ice. Although HAdV-B3 did not show any infection of HCE cells in the titration assay, it was still used in the infection assay (1000 vp/cell). To remove unbound viruses, cells were washed three times with serum-free growth medium. The cells were incubated for 44 h at 37 °C in culture medium containing 1% FBS. After 44 h incubation, the cells were washed once with PBS (pH 7.4) and fixed with ice-cold methanol. The cells were then incubated with monoclonal mouse anti-adenovirus antibody (1:250) diluted in PBS for 1 h at room temperature (RT). The cells were washed three times with PBS and incubated for 1 h at RT with donkey anti-mouse Alexa Fluor 488 antibody (1:1000) diluted in PBS. The cells were washed once with PBS and stained with 4′,6-diamidino-2-phenylindole (DAPI; diluted 1:5000 in PBS) for 5 min. The cells were then washed twice with PBS. The infection of cells was analyzed in Trophos (Luminy Biotech, Marseille, France). Infection assays were performed with the following variations: (i) HCE cells were treated with 20 mU/mL of neuraminidase for 1 h at 37 °C before incubating with the viruses and (ii) viruses were incubated with ME0462 (diluted in serum-free growth medium) for 1 h at 4 °C on ice before incubating with cells. Untreated cells and viruses were used as control.

### 2.4. Cytopathic Effect Analysis

A549 and HCE cells were grown as a monolayer (90% confluency) in 25 cm^2^ flasks. Cells were incubated with HAdV-B3 virus (1:100 dilution) for 1 h at 37 °C. After 1 h, cells were washed with serum-free growth media and fresh media containing 10% FBS were added to cells. The cells were incubated for 36 h at 37 °C. Virus-induced cytopathic effect was observed under the light microscope (Zeiss, Oberkochen, Germany). 

### 2.5. Virus Cell-Binding Assays

HCE cells were detached with pre-warmed PBS containing 0.05% ethylenediaminetetraacetic acid (EDTA). The cells were counted and reactivated in 10% growth medium for 1 h at 37 °C (in suspension). The cells were then pelleted in a V-bottom 96 well plate (1 × 10^5^ cells/well) and washed once with binding buffer (BB; DMEM supplemented with 20 mM HEPES, PEST, and 1% bovine serum albumin). ^35^S-labeled HAdV-D37, -D53, and -D64 viruses (10000 vp/cell, diluted in BB) were added to cells and incubated for 1 h at 4 °C on ice. To remove unbound viruses, cells were washed three times with BB. Cell-associated radioactivity was measured by using Wallac 1409 liquid scintillation counter (Perkin-Elmer, Waltham, MA, USA). The assay was performed with the following variations: (i) HCE cells were treated with 20 mU/mL of neuraminidase for 1 h at 37 °C before incubating with viruses and (ii) viruses were incubated with increasing concentrations of ME0462 (diluted in BB) for 1 h at 4 °C before incubating with HCE cells. Untreated cells and viruses were used as control.

### 2.6. Fiber Knob Binding Assays

HCE cells were detached with pre-warmed PBS containing 0.05% EDTA. The cells were counted and then reactivated in 10% growth medium for 1 h at 37 °C (in suspension). After 1 h, cells (2 × 10^5^ cells/well) were pelleted in a V-bottom shaped 96 well plate and washed once with BB. The cells were then incubated with 10 μg/mL of fiber knobs in 100 μL BB for 1 h at 4 °C on ice. Unbound fiber knobs were washed away with BB. The cells were then incubated with monoclonal mouse anti-RGS-His antibody (diluted 1:200 in BB) for 1 h at 4 °C on ice. After 1 h of incubation, the cells were washed once with BB and incubated with monoclonal donkey anti-mouse Alexa Fluor 488 antibody (diluted 1:1000 in BB) for 1 h at 4 °C on ice. Thereafter, the cells were washed with flow cytometry (FACS) buffer (PBS with 2% FBS) and analyzed by flow cytometry using a FACS LSRII instrument (Becton Dickinson, Franklin Lakes, NJ, USA). The results were analyzed using FACSDiva software (Becton Dickinson, Franklin Lakes, NJ, USA). The assay was performed with the following variations: (i) HCE cells were treated with 20 mU/mL of neuraminidase for 1 h at 37 °C before incubating with the fiber knobs and (ii) fiber knobs were incubated with ME0462 (diluted in BB) for 1 h at 4 °C before incubating with HCE cells. Untreated cells and fiber knobs were used as control.

### 2.7. Synthesis of SA-Based Inhibitor ME0462 

Synthesis of ME0462 was performed as reported previously [28].

### 2.8. Surface Plasmon Resonance (SPR) Analysis 

SPR was performed in a Biacore T200 instrument (GE, Chicago, IL, USA). Fiber knobs were diluted in running buffer (0.01 M HEPES pH 7.4, 0.15 M NaCl, 100 µM EDTA, 0.005% *v*/*v* Surfactant P20) to a concentration of around 0.035 μM (0.034–0.047 μM) and immobilized on the Ni-NTA sensor chip (GE Healthcare, Chicago, IL, USA) according to the manufacturer’s instructions, resulting in an immobilization density between 1100 and 1500 response unit (RU). In short: An automated program cycle of the following sequence: (1) Activation of the sensor chip with Ni(II), (2) Capture of fiber knobs (3) Analyte injection, (4) Regeneration of the surface with 0.3 M EDTA, and (5) Running buffer without EDTA. All steps were performed at a flow rate of 30 μL/min. All assays were carried out at 25 °C. The analyte (ME0462) were serially diluted in running buffer to prepare a two-fold concentration series ranging from 40–0,3125 μM and then injected in series over the reference and experimental biosensor surfaces for 60 s and a dissociation time of 60 s. Blank samples containing only running buffer were also injected under the same conditions to allow for double referencing. The binding affinities (KDs) were calculated using BIAcore T200 evaluation software (GE, Chicago, IL, USA).

### 2.9. Statistical Analysis

All experiments were performed two or three times with duplicate or triplicate samples. All results are presented as standard error of mean (SEM). Graphical and statistical analyses were performed by using GraphPad Prism version 7 for Windows (GraphPad Software). Significance was calculated using Student’s *t*-test. All *p*-values of <0.05 were considered statistically significant.

## 3. Results and Discussion

HAdVs utilize the knob domain of the fiber protein for the attachment to cellular receptors [20,31]. X-ray crystallographic analyses revealed that SA binds to the positively charged central cavity on the fiber knob of EKC-causing HAdV-D37 and that Tyr312, Pro317, and Lys345 are critical for SA-interactions [25,30]. Thr310 and Ser344 residues provide additional contacts via water-mediated hydrogen bonds. To investigate whether SA-interacting residues are conserved among ocular HAdVs, we analyzed the homology of the fiber knobs of ocular HAdVs. We found that in respect to HAdV-D37 fiber knob, the residues that make direct contact with SA are entirely conserved (highlighted in cyan) (Figure 1) among all EKC-causing HAdVs, whereas other residues are partially conserved (highlighted in yellow). This suggests that all EKC-causing HAdVs may bind to SAs and potentially use SA-containing glycans as cellular receptors. PCF-causing HAdVs only contained Tyr312 (in respect to HAdV-D37 fiber knob) as conserved SA-binding residue and this may not favor the binding of SAs to the fiber knobs of these viruses. However, we could not exclude the possibility of HAdV-B3 and -E4 fiber knobs binding to SA with a different mode of action. In support of this, we have recently shown that HAdV-G52 short fiber knob binds to SA-containing glycans by engaging a charged steering rim and different sets of residues [32,33]. Moreover, lysine or alanine at 240 in the fiber knobs of EKC-causing HAdVs has been proposed as signature residue (highlighted in brown), which distinguish EKC from non-EKC-causing HAdVs and contribute in determining the corneal tropism of EKC-causing HAdVs [34]. Notably, HAdV-B3 also contains lysine residue at 240 with respect to the fiber knobs of EKC-causing species D HAdVs.

To investigate whether SA is needed for the infection of HCE cells by ocular HAdVs, we analyzed the infection of HAdV-B3, -E4, -D37, -D53, -D56, and -D64 in HCE cells pre-treated with neuraminidase, which removes SAs from the cell-surface. Pre-treatment of HCE cells with neuraminidase reduced the infection of cells by HAdV-D53 (~85%) and -D64 (~90%), which was similar to the reduction of infection by HAdV-D37 (~90%) (Figure 2A). This indicates the crucial function of cell-surface SAs for these viruses. Neuraminidase treatment did not alter HAdV-D56 infection of HCE cells, which is interesting since this virus has a conserved SA-binding site in its fiber knob. However, this result is in agreement with a previous report, which also demonstrated that HAdV-D56 does not use SAs as receptors on HCE cells [35]. This highlights that HAdV-D56 has evolved to utilize an unknown, SA-independent infection/entry mechanism. Onwards in this study, if not mentioned, the term EKC-causing HAdVs will represent the neuraminidase-sensitive EKC-causing HAdVs, not including HAdV-D56. Unfortunately, we were unable to propagate HAdV-D8 and -D54 viruses, thus, these viruses could not be included in this study. We observed that the infection of HCE cells by HAdV-E4 was independent of cell-surface SAs. HAdV-E4 uses CAR as cellular receptors [22] and HCE cells do express CAR [32]. The ability of HAdV-E4 to infect HCE cells may explain its corneal tropism. Surprisingly, we did not observe any infection of HCE cells by HAdV-B3. HAdV-B3 uses desmoglein-2 (DSG-2) as cellular receptors on epithelial cells of different tissue origins [21]. DSG-2 is an integral part of desmosomes and highly expressed on epithelial cells [36]. As to our knowledge, the expression of DSG-2 on human corneal tissue and on HCE cells has not been reported. Studies have shown DSG-2 expression on the epithelium of rodent and bovine corneas [37,38] and given its central role in cell-cell adhesion of epithelial cells, it is likely that HCE cells also express DSG-2. This highlights that HAdV-B3 may rely on other cellular factors for the infection of HCE cells. We also confirmed the infectious nature of HAdV-B3 by analyzing the cytopathic effect (CPE) of HAdV-B3 on human lung epithelial carcinoma cells (A549) and HCE cells. HAdV-B3 displayed CPE on A549 cells but not on HCE cells after 36 h post infection (Figure 2B). This is an interesting observation and may be related to the restricted conjunctival tropism of HAdV-B3.

To further characterize SAs as cellular receptors for neuraminidase-sensitive EKC-causing HAdVs, we first examined the binding of ^35^S-labelled HAdV-D37, -D53, and -D64 to HCE cells pre-treated with neuraminidase. Neuraminidase treatment significantly reduced the binding of these viruses to cells (Figure 3A), which confirms that these HAdVs require cell-surface SAs for efficient attachment to HCE cells. HAdV-D37 fiber knob contains an overall positive charge (isoelectric point = 9.14) and has a highly positively charged SA-binding central cavity on the top of the fiber knob [26,30]. HAdV-D8, -D53, -D54, and -D64 fiber knobs also have a high isoelectric point [39] (Table 1), thus, we assume that fiber knobs of these viruses can form a similar positively charged central cavity, which can accommodate cell-surface SAs. EKC-causing HAdV fiber knobs for HAdV-D8 and -D37 are 100% identical to those for HAdV-D53 and -D64, respectively (Table 1). In this study, HAdV-D8 fiber knob is used as a representative for HAdV-D53 fiber knob and denoted as HAdV-D8/53 and HAdV-D37 fiber knob is used as a representative for HAdV-D64 fiber knob and denoted as HAdV-D37/64. Notably, HAdV-D54 fiber knob is highly similar (>97% identical) to the fiber knob of HAdV-D8 [6]. Thus, it is likely that the fiber knobs of these viruses also bind to cellular-surface SAs. To demonstrate the capacity of the fiber knobs of EKC-causing HAdVs to bind to cell-surface SAs, we analyzed the binding of the fiber knobs to HCE cells pre-treated with neuraminidase. Neuraminidase treatment significantly reduced the binding of fiber knobs of all EKC-causing HAdVs (Figure 3B). Taken together, these data provide substantial evidence that SA-containing glycans function as cellular receptors not only for HAdV-D37 but also for HAdV-D53 and -D64 on HCE cells, and that the interaction is mediated by fiber knobs. Importantly, the reduced binding of HAdV-D54 fiber knobs to neuraminidase-treated HCE cells indicates that HAdV-D54 virus can potentially use SA-glycans as cellular receptors on these cells. However, we emphasize that the receptor usage of HAdV-D8 and -D54 on HCE cells must be investigated by using whole virions. We want to highlight that HAdV-D37 fiber knobs also bind to HCE cells via charge-dependent mechanisms to sulfated glycosaminoglycans (GAGs), where GAGs function as decoy receptors [40]. As discussed earlier, all EKC-causing HAdVs contain highly positively charged fiber knobs, therefore, the remaining residual binding observed in these results may be the outcome of the binding of these viruses and their fiber knobs to cell-surface sulfated GAGs.

Targeting the attachment of viral pathogens to their cellular receptors is an attractive approach for the development of antiviral drugs. This approach offers several advantages; first, these inhibitors block the very first step of the viral infection cycle, which halts the viral replication and progeny virus production. Second, these inhibitions act on the extracellular level, which minimizes the risks of off-target effects on intracellular factors. HAdV-D37 fiber knob contains three SA-binding sites having one SA-binding site in each monomer [30]. Recently, a trivalent SA derivative i.e., ME0462 was found as a highly potent inhibitor of HAdV-D37 binding to and infection of HCE cells [28,41]. SA moieties of ME0462 mediate direct contact with critical SA-binding residues on HAdV-D37 fiber knob, which prevents the virus attachment to its SA-containing receptors and consequently inhibits virus infection of cells [28]. Given that HAdV-D53 and -D64 also contain critical SA-binding residues and use SAs as cellular receptors, we assumed that ME0462 may also inhibit HAdV-D53 and -D64 binding to and infection of HCE cells. To test this, we incubated ^35^S-labelled HAdV-D37 (used as a control), -D53 and -D64 virions with increasing concentrations of ME0462 and analyzed virus binding to HCE cells [28]. Interestingly, ME0462 efficiently inhibited both HAdV-D53 and -D64 binding to cells with IC50 values of 218 nM and 3.06 nM, respectively (Figure 4A). As expected, ME0462 also inhibited HAdV-D37 binding to cells with a low nM IC50 value (2.78 nM), which is in agreement with previous results [28]. This data indicates that ME0462 holds the capacity to inhibit the binding of multiple EKC-causing HAdVs through a similar mechanism as reported for HAdV-D37. Further, to elucidate the ability of ME0462 to inhibit virus infection, we pre-incubated HAdV-D37, -D53, and -D64 with ME0462 and analyzed virus infection of HCE cells. Indeed, ME0462 inhibited HAdV-D37, -D53, and -D64 infection of cells (Figure 4B). Taken together, these results clearly suggest that ME0462 can certainly be used as an antiviral against three major causative agents of EKC. We further wanted to confirm whether ME0462 prevents HAdV-D37, -D53, and -D64 binding and infection by interfering with the interaction between the viral fiber knobs and cellular SA-receptors. To demonstrate this, we pre-incubated HAdV-D8/53, -D37/64, and -D54 fiber knobs with ME0462 and analyzed the binding of knobs to HCE cells. ME0462 significantly inhibited binding of HAdV-D8/53 and -D54 fiber knobs by 60% to cells (Figure 4C), which shows that ME0462 indeed disrupts the interactions between viruses (via fiber knobs) and their SA-containing receptors. As expected, ME0462 also inhibited (by 70%) HAdV-D37/64 (used as a control) fiber knob binding to cells. Importantly, inhibition of HAdV-D54 fiber knobs by ME0462 suggests that ME0462 can also be considered as antiviral against emerging EKC-causing HAdV-D54. The usage of ME0462 as antiviral has promising advantages i.e., it would not require systemic administration and could be applied topically (e.g., as eye drops). This will overcome the poor pharmacokinetic profiles related to glycan-based drugs such as rapid clearance from serum and low cellular uptake [42]. Additionally, ocular toxicity of ME0462 has also been investigated in rabbits without any sign of toxicity and/or adverse effects [28]. Further, to confirm a direct interaction and to determine the affinity between HAdV-D8/53, -D37/64, and -D54 fiber knobs and ME0462, we performed SPR analysis. ME0462 (in solution) interacted with all fiber knobs (immobilized) with low micromolar affinities (Table 2), demonstrating the ability of ME0462 to directly bind to the fiber knobs assumingly by engaging critical SA-binding residues/cavity.

In summary, we conclude that SA-containing glycans function as cellular receptors for five out of six EKC-causing species D HAdVs that cause EKC on a regular basis. It appears that cell-surface SA has benefited ocular viruses to establish corneal infections and this may have led to the positive selection of SAs as cellular receptors during the evolution of these viruses. HAdV-B3 and -E4 have previously been shown to use DSG-2 and CAR as receptors on cells of non-ocular origins, respectively. The latter two viruses either completely failed to infect HCE cells (HAdV-B3), or infect these cells or cause a less severe form of EKC (HAdV-E4). Our results point out that there is a pronounced correlation between receptor usage and tropism of the ocular HAdVs investigated in this study. We also demonstrate that SA-containing molecule i.e., ME0462, which is designed to bind with high affinity to the SA-binding sites in the viral fiber knob, efficiently protect HCE cells from the infection by multiple EKC-causing HAdVs. We also highlight that a different antiviral development approach should be applied to combat ocular infections caused by ocular HAdVs that do not use SA-containing glycans as cellular receptors.

## Figures and Tables

**Figure 1 viruses-11-00395-f001:**
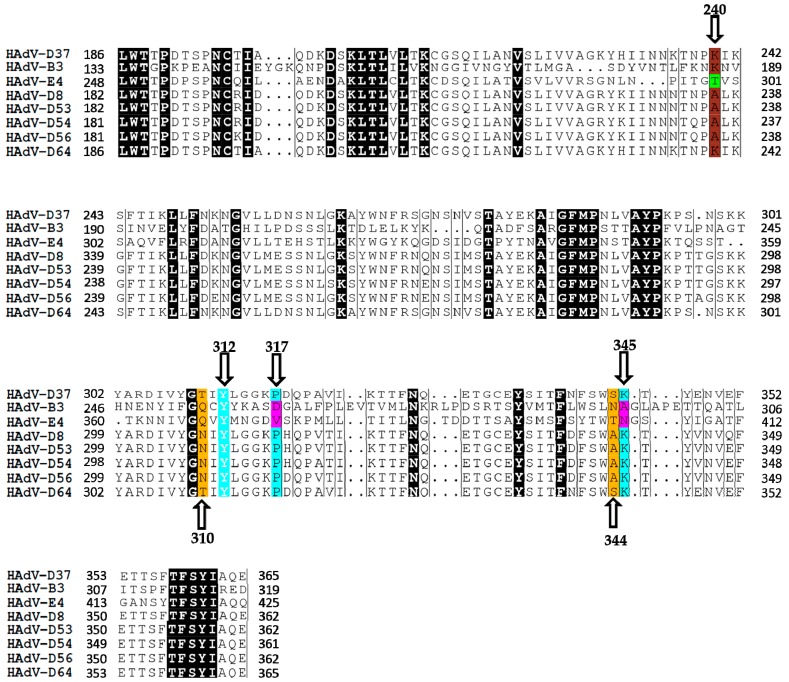
Multiple sequence alignment of the fiber knobs of epidemic keratoconjunctivitis (EKC)- and pharyngoconjunctival fever (PCF)- causing HAdVs. EKC- and PCF-causing HAdVs contain highly and partially conserved critical sialic acid (SA)-binding residues, respectively. Residues highlighted in cyan with white texts represent conserved SA-binding residue (Tyr312) among all ocular HAdVs. Residues highlighted in cyan with black texts represent conserved SA-binding residues (Pro317 and Lys345) among all EKC-causing HAdVs fiber knobs and these residues are not conserved (highlighted in pink) in HAdV-B3 and -E4 fiber knobs. Residues at 310 and 344 on HAdV-D37 mediate contacts with SA via water-mediated hydrogen bonds and these residues either partially or not conserved among all ocular HAdVs (highlighted in yellow). The crucial residue (lysine or alanine) at 240 is conserved (highlighted in brown) in the fiber knobs of all ocular HAdVs except for HAdV-E4 (highlighted in green). Amino acid start positions are according to the HAdV-D37 sequence (accession No. DQ900900).

**Figure 2 viruses-11-00395-f002:**
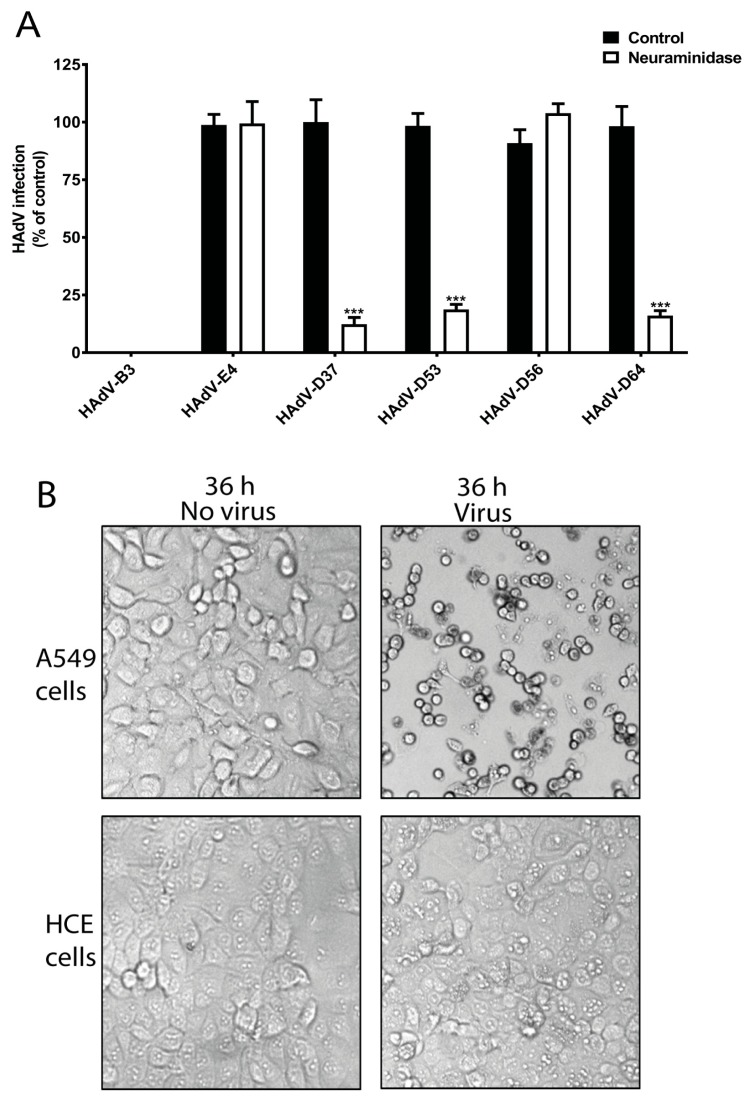
Infection of ocular HAdVs on neuraminidase pre-treated human corneal epithelial (HCE) cells. (**A**) HAdV-B3, -E4, -D37, -D53, -D56, and –D64 infection of HCE cells pre-treated with and without neuraminidase. (**B**) Cytopathic effect (CPE) analysis of HAdV-B3 on A549 and HCE cells. Error bars represent mean ± SEM. *** *p* < 0.001 relative to control.

**Figure 3 viruses-11-00395-f003:**
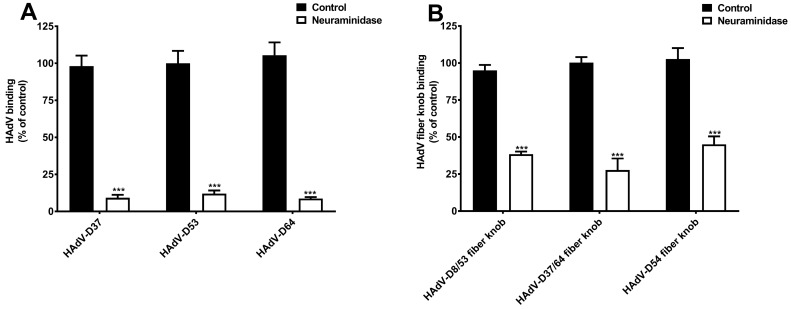
Effect of neuraminidase treatment on the binding of epidemic keratoconjunctivitis (EKC)- causing HAdVs to human corneal epithelial (HCE) cells. (**A**) Binding of ^35^S-labelled HAdV-D37, -D53, and -D64 to HCE cells pre-treated with and without neuraminidase. (**B**) Binding of HAdV-D8/53, -D37/64, and -D54 fiber knobs to HCE cells pre-treated with and without neuraminidase. Error bars represent mean ± SEM. *** *p* < 0.001 relative to control.

**Figure 4 viruses-11-00395-f004:**
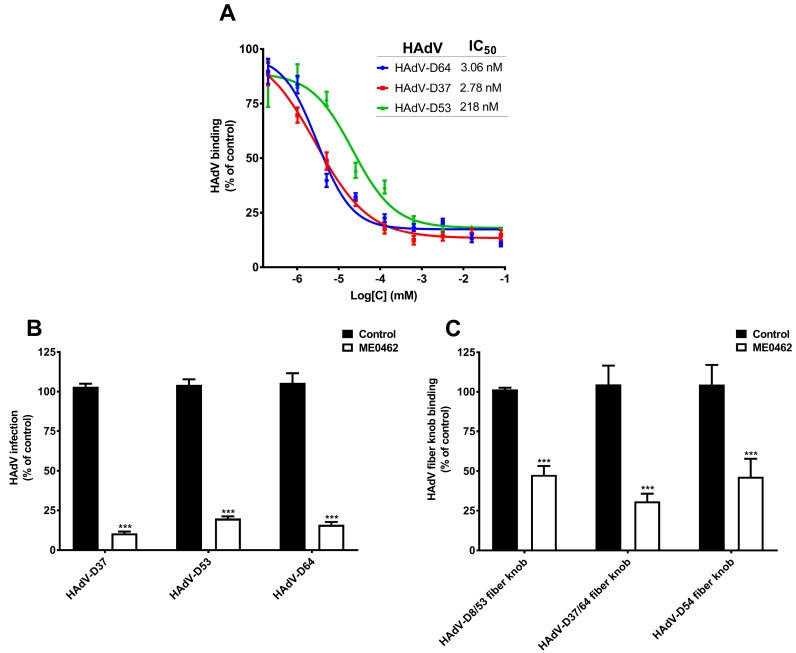
Effect of ME0462 on epidemic keratoconjunctivitis (EKC)-causing HAdVs binding to and infection of human corneal epithelial (HCE) cells. (**A**) Binding of ^35^S-labeled HAdV-D37, -D53, and -D64, pre-incubated with increasing dose-dependent concentrations of ME0462, to HCE cells. (**B**) HAdV-D37, -D53, and -D64, pre-incubated with ME0462 (80 µM), infection of HCE cells. (**C**) HAdV-D8/53, -D37/64, and -D54 fiber knobs, pre-incubated with ME0462 (80 µM), binding to HCE cells. Error bars represent mean ± SEM. *** *p* < 0.001 relative to control.

**Table 1 viruses-11-00395-t001:** Amino acid sequence identities of the fiber knobs of epidemic keratoconjunctivitis (EKC)-causing HAdVs and their isoelectric points (pI).

HAdV Fiber Knobs	Identical/Homologues Fiber Knobs	% Identity	Isoelectric Points (pI)
HAdV-D8	HAdV-D53	100	9.04
HAdV-D37	HAdV-D64	100	9.14
HAdV-D54	HAdV-D8	>97	8.88

**Table 2 viruses-11-00395-t002:** Surface plasmon resonance (SPR) analysis of HAdV fiber knob interactions with ME0462.

Analyte (In Solution)	HAdV Fiber Knobs (Immobilized)	K_D_ (µM) ± SD
ME0462	HAdV-D8/53	26.1 ± 5.93
	HAdV-D37/64	14.2 ± 1.97
	HAdV-D54	≥40 ± 0.00

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
