# Peer review of "Sialic Acid-Containing Glycans as Cellular Receptors for Ocular Human Adenoviruses: Implications for Tropism and Treatment"

_viruses, 2019, doi:10.3390/v11050395_

Round 1
Reviewer 1 Report
The presented findings in this study appear to be of high quality with robust data as could be expected from the Arnberg team; experts in adenoviral binding to cellular receptors with a focus on HAdV types D, B and E. In this manuscript the authors further dissect the mechanisms of entry of type D viruses that cause epidemic keratoconjunctivitis. They identify HAdV-D53 and HAdV-D64 as dependent on sialic acid cell surface binding in human corneal cells in culture. The positive control is their previously characterised HAdV-D37 binding to sialic acid and the negative controls HAdV-B3 (does not infect corneal cells) and HAdV-E4 (infects corneal cells but is not sialic acid dependent).
The study rationale and findings are logically outlined and the conclusions based on different approaches, validating the data. A variety of methods were used for example, infection assays, 35S-labelled virus for cell binding assays, generation of fibre proteins for binding assays, plasmon resonance and competition with the previously characterised synthetic molecule ME0462 that specifically binds to the sialic acid binding sequences in HAdV-D37.
My only suggestion for improvement is the following: for readers less knowledgable in adenoviral fibre structure it would be helpful to include the sequence region (e.g. nucleotide numebers and locartion in fibre) in Figure 1. If possible, a theoretical illustration suggesting how the sialic acid binds to this domain.
This manuscript is clearly very interesting, important, of high quality and is therefore recommended for publication.
Author Response
We thank this expert for reviewing our manuscript and giving positive comments. Below is our response to the only suggestion given by this reviewer.
The presented findings in this study appear to be of high quality with robust data as could be expected from the Arnberg team; experts in adenoviral binding to cellular receptors with a focus on HAdV types D, B and E. In this manuscript the authors further dissect the mechanisms of entry of type D viruses that cause epidemic keratoconjunctivitis. They identify HAdV-D53 and HAdV-D64 as dependent on sialic acid cell surface binding in human corneal cells in culture. The positive control is their previously characterised HAdV-D37 binding to sialic acid and the negative controls HAdV-B3 (does not infect corneal cells) and HAdV-E4 (infects corneal cells but is not sialic acid dependent).
The study rationale and findings are logically outlined and the conclusions based on different approaches, validating the data. A variety of methods were used for example, infection assays, 35S-labelled virus for cell binding assays, generation of fibre proteins for binding assays, plasmon resonance and competition with the previously characterised synthetic molecule ME0462 that specifically binds to the sialic acid binding sequences in HAdV-D37.
Point 1: My only suggestion for improvement is the following: for readers less knowledgable in adenoviral fibre structure it would be helpful to include the sequence region (e.g. nucleotide numebers and locartion in fibre) in Figure 1. If possible, a theoretical illustration suggesting how the sialic acid binds to this domain.
Author’s response- We agree with the reviewer. Now, we added the corresponding amino acid (not nucleotide) numbers in the sequences (Figure 1). However, HAdV-D37 fiber knob interaction with sialic acid and other sialic acid-containing molecules have been characterized by X-ray crystallography and published previously, thus we prefer to cite articles rather providing the illustration.
This manuscript is clearly very interesting, important, of high quality and is therefore recommended for publication.
Reviewer 2 Report
The manuscript by Chandra et al., is well presented, covers a trending area of adenovirus research and most importantly - presents novel data important to understand the pathogenic ocular HAdVs. I would specifically point out the well-written, easy to understand and interesting introduction part. The experiments clearly show that in addition to classical HAdV-D37 also 2 additional HAdVs (D53 and D64) use SA for binding to the HCE cells. Further, the finding that ME0462 blocks binding of D53 and D64 to HCE, highlights the usefulness of the ME0462 as an inhibitor of at least 3 ocular HAdVs (D37, D53, D64).
Although the manuscript is well written and data nicely presented, I still have a few small comments, which the authors may take into consideration:
Line 27: "...infection of multiple EKC- causing..." I feel that in this context the word "multiple" is an overstatement since the ME0462 was tested only on 3 ocular viruses (and the data for D37 was know before). Consider rephrasing the sentence.
Line 104: -D64 is missing from the virus list. Further, it remains unclear to me if different virus types were titrated or not (vp or PFU). This can be important when doing infection experiments (Fig. 1), where comparing the effect caused by different virus types can be influenced by the amount (either vp of PFU) of used virus. Please, state if viruses were titrated and how much of the virus (vp or PFU) were used for the infection exp. (Fig.1).
Lines 236-237: "HAdv-B3 displayed CPE on A549 cells but not on HCE cells after 36hpi.This may perhaps be one of the reasons for restricted conjunctival tropism of HAdV-B3." The lack of visual CPE can not be only due to the restricted tropism. For example the lack of CPE can be also due to the malfunctioning of the adenovirus death protein (Tollefson., 1996). Hence, I would suggest to rephrase the sentence (line 237).
Fig. 1: In line with my previous comment, could it be that one reason why B3 is not "infecting" the HCE cells is just technical issue. Simply, the used antibody (Mab8052) is just not recognising the B3, but detects well the other viruses (D37, D56, D64) in the infected HCE cells?
Fig. 4B &4C: it remains unclear to me what is the label "TSA" stands for? Should not it be "ME0462" instead?
Author Response
We thank the reviewer for reviewing our manuscript and for providing constructive comments and suggestions. Below are our responses point-by-point to the specific comments.
The manuscript by Chandra et al., is well presented, covers a trending area of adenovirus research and most importantly - presents novel data important to understand the pathogenic ocular HAdVs. I would specifically point out the well-written, easy to understand and interesting introduction part. The experiments clearly show that in addition to classical HAdV-D37 also 2 additional HAdVs (D53 and D64) use SA for binding to the HCE cells. Further, the finding that ME0462 blocks binding of D53 and D64 to HCE, highlights the usefulness of the ME0462 as an inhibitor of at least 3 ocular HAdVs (D37, D53, D64).
Although the manuscript is well written and data nicely presented, I still have a few small comments, which the authors may take into consideration:
Point 1. Line 27: "...infection of multiple EKC- causing..." I feel that in this context the word "multiple" is an overstatement since the ME0462 was tested only on 3 ocular viruses (and the data for D37 was known before). Consider rephrasing the sentence.
Author’s response: We agree, and modified the text accordingly.
Point 2. Line 104: -D64 is missing from the virus list. Further, it remains unclear to me if different virus types were titrated or not (vp or PFU). This can be important when doing infection experiments (Fig. 1), where comparing the effect caused by different virus types can be influenced by the amount (either vp of PFU) of used virus. Please, state if viruses were titrated and how much of the virus (vp or PFU) were used for the infection exp. (Fig.1).
Author’s response: Thanks for noticing the error. D64 is added now (line 105). Yes, viruses were titrated before the infection assay. As per reviewer’s suggestion the following sentence is added:
“Prior to infection experiment, we titrated the virus stocks, and used dilutions resulting in infection of approximately 5 % of cells in each well. HAdV-E4 (3000 vp/cell), -D37 (700 vp/cell), -D53 (550 vp/cell), -D56 (500 vp/cell), and -D64 (300 vp/cell)” were added to cells and incubated for 1 h on ice”. Although HAdV-B3 did not show any infection of HCE cells in the titration assay, it was still used in the infection assay (1000 vp/cell).” (line 127-132).
Point 3. Lines 236-237: "HAdv-B3 displayed CPE on A549 cells but not on HCE cells after 36hpi. This may perhaps be one of the reasons for restricted conjunctival tropism of HAdV-B3." The lack of visual CPE cannot be only due to the restricted tropism. For example the lack of CPE can be also due to the malfunctioning of the adenovirus death protein (Tollefson., 1996). Hence, I would suggest to rephrase the sentence (line 237).
Authors’ response: We agree to this reviewer and we have modified the text accordingly (line 241-242).
Point 4. Fig. 1: In line with my previous comment, could it be that one reason why B3 is not "infecting" the HCE cells is just technical issue. Simply, the used antibody (Mab8052) is just not recognising the B3, but detects well the other viruses (D37, D56, D64) in the infected HCE cells?
Author’s response: Mab8052 is a monoclonal antibody, which is raised against HAdV-B3 as the immunogen, thus it is very unlikely that Mab8052 would not recognize HAdV-B3. We have also used rabbit polyclonal sera raised against HAdV-B3 and found that HAdV-B3 cannot infect HCE cells, whereas A549 cells are efficiently infected.
Point 5. Fig. 4B &4C: it remains unclear to me what is the label "TSA" stands for? Should not it be "ME0462" instead?
Author’s response: Thanks for noticing the error (now corrected).